# A New Species of the Pythonomorph *Carentonosaurus* from the Cenomanian of Algora (Guadalajara, Central Spain) [note 1]

**DOI:** 10.3390/ani13071197

**Published:** 2023-03-29

**Authors:** Alberto Cabezuelo-Hernández, Adán Pérez-García

**Affiliations:** Grupo de Biología Evolutiva, Departamento de Física Matemática y de Fluidos, Facultad de Ciencias, UNED, Avda. Esparta s/n, Las Rozas, 28232 Madrid, Spain; a.perez.garcia@ccia.uned.es

**Keywords:** Southwestern Europe, Iberian Peninsula, Late Cretaceous, Pythonomorpha, *Carentonosaurus algorensis* sp. nov., pachyosteosclerosis

## Abstract

**Simple Summary:**

Isolated bony material is common among fossil remains, usually hindering the taxonomic identification of these elements. Vertebrae are among the most abundant isolated remains from squamate ‘lizards’; however, these elements are highly diagnostic within this group, so that they are usually recognized as adequate to make taxonomic determinations. In this context, the anatomy and inner osseous morphology of an isolated ‘lizard’ vertebra from the early Late Cretaceous (circa 95 m.a) of Spain is here described in detail. Vertebral anatomy is of high interest when studying the ecology and the evolutionary adaptations of a particular taxon, as it provides valuable information about the main patterns and constraints associated with locomotion. In this context, functional and paleoecological implications for the taxon studied here are provided based on its external and inner osseous features. The detailed study of the vertebral material here presented allowed us to identify a new species of aquatic squamate ‘lizard’. This new taxon displays adaptations to shallow marine environments, evidencing the high diversification of these forms during the early Late Cretaceous of Europe.

**Abstract:**

The Cenomanian (lowermost Upper Cretaceous) faunal assemblages are of high interest in understanding the turnovers that took place between the Early and the Late Cretaceous, resulting in significant differences. In this context, the analysis of the association of reptiles found in the Algora fossil site (Guadalajara Province, Central Spain) is of great interest since it represents the first European Cenomanian site with a high concentration of macrovertebrate remains. A new pythonomorph ‘lizard’ from Algora, *Carentonosaurus algorensis* sp. nov., is described here. It is the second representative of this European genus. Its microanatomical study reveals that an extreme pachyosteosclerosis affected at least its dorsal vertebrae, suggesting adaptations for slow-swimming habits in shallow-water environments. Consequently, this new taxon is interpreted as a slow swimmer, hovering near the bottom of near-shore marine environments of the Late Cretaceous European Archipelago and, more specifically, along the shores of the larger Iberian Island for that period. This is in concordance with the high diversification of ‘pachyostotic’ pythonomorphs recorded during the Cenomanian, allowing the subsequent adaptation of this lineage to open marine environments.

## 1. Introduction

The Cenomanian (lowermost Late Cretaceous) is a key stage to analyze the faunal turnover evidenced between the faunas of the Lower Cretaceous and those of the uppermost Cretaceous [1]. In this context, the Spanish Algora fossil site (Guadalajara Province, Central Spain; Figure 1A), deposited at the uppermost middle to lowermost upper Cenomanian, represents the first European site with a high concentration of macrovertebrate remains from the Cenomanian (see Pérez-García et al. [2] and references therein). Thus, the relatively poorly known faunal assemblages found in the Cenomanian of this continent, whose knowledge has been significantly improved thanks to the finds in this Spanish site, are currently identified as more akin to those found in the uppermost Cretaceous than to those of the Early Cretaceous.

The documented vertebrate faunal assemblage in the Algora Cenomanian site is represented by an osteichthyan member corresponding to *Obaichthys africanus* Grande, 2010 [3]; a stem turtle attributable to the helochelydrid aff. *Plastremys lata* Parkinson, 1881 [4]; a pleurodiran bothremydid turtle described based on remains from this locality, *Algorachelus peregrina* Pérez-García, 2017 [5]; an indeterminate elasmosaurian sauropterygian; two neosuchian Crocodyliformes, of which one corresponds to a member of Eusuchia and the other to a most basal form; a theropod probably belonging to Abelisauridae; and a so far undescribed new lithostrotian sauropod (see Pérez-García et al. [2] and references therein). In this context, the faunal list of the site is increased here. Thus, a squamate remain is identified, representing the first record of this clade in the site and increasing the European record of marine pythonomorphs (Figure 1B).

The oldest European squamates are known from the Bathonian (Middle Jurassic) of Britain [6,7], whereas the oldest Iberian squamates are from the Kimmeridgian (Late Jurassic) of Portugal [8]. The Iberian Lower Cretaceous squamate assemblage is represented by a mixture of relict Jurassic and more derived forms exclusively from this period. Most Iberian Upper Cretaceous squamates correspond to other lineages that reached this continent during the Cenomanian or posteriorly [8]. In this sense, a radiation of aquatic squamates (i.e., that of the Pythonomorpha sensu Lee [9]) occurred during the early Cenomanian [7]. The pythonomorphs quickly diversified, mainly in the Tethys region and, to a lesser extent, in the Western Interior Sea [1,10], originating the ‘hind-limbed snakes’ (‘pachyophiids’), ‘dolichosaurs’, ‘aigialosaurs’, and mosasaurids, all these groups having been recorded in the Iberian Peninsula (e.g., [11,12]). The mosasaurids, unlike the other groups, became successful in the later Cretaceous, both paleobiogeographically (as they were distributed worldwide) and in diversity [1,13]. During the Cenomanian, several groups of Pythonomorpha developed into ‘pachyostotic’ forms [14,15], among which both osteosclerosic and pachyosteosclerosic (see Houssaye [16,17] for definitions) taxa were present, already indicating diverse aquatic adaptations within the first radiation of marine pythonomorphs [1,10,18].

The aim of this paper is the detailed description and systematic analysis of the squamate remain found in the Cenomanian of Algora, the presence of this clade in the site not having been reported until now. Moreover, paleoecological and functional implications regarding the bone hypertrophy of this taxon are discussed.

*Anatomical abbreviations*. *cb*, cancellous bone; *cd*, condyle; *ct*, cotyle; *ir*, interzygapophyseal ridge; *itc*, interzygapophyseal constriction; *lf*, lateral foramen; *nc*, neural canal; *ns*, neural spine; *pb*, periosteal bone; *pcf*, paracotylar foramen; *pd*, paradiapophysis; *prf*, prezygapophyseal facet; *prz*, prezygapophysis; *ptf*, postzygapophyseal facet; *ptz*, postzygapophysis; *pzf*, parazygosphenal foramen; *sg*, spinal grooves; *sb*, subcentral border; *sbf*, subcentral foramen; *zs*, zygosphene; zsf, zygosphenal facets; *zl*, zygantral lamina; *zt*, zygantrum; *ztf*, zygantral facets.

*Institutional abbreviations*. ALG, Algora collection, Museo de Paleontología de Castilla-La Mancha, Cuenca, Spain; EJ, ‘Ein Yabrud collections of the The Hebrew University, Jerusalem, Israel; HUJ-PAL, Hebrew University of Jerusalem, Paleontological Collections, Jerusalem, Israel; MA MAD, “Île Madame” collection of the Musée d’Angoulême, Angoulême, France; MNCN, Museo Nacional de Ciencias Naturales, Madrid, Spain; MNHN IMD, “Île Madame” collection of the Muséum National d’Histoire Naturelle, Paris, France; UR1, Université de Rennes 1, Rennes, France.

**Figure 1 animals-13-01197-f001:**
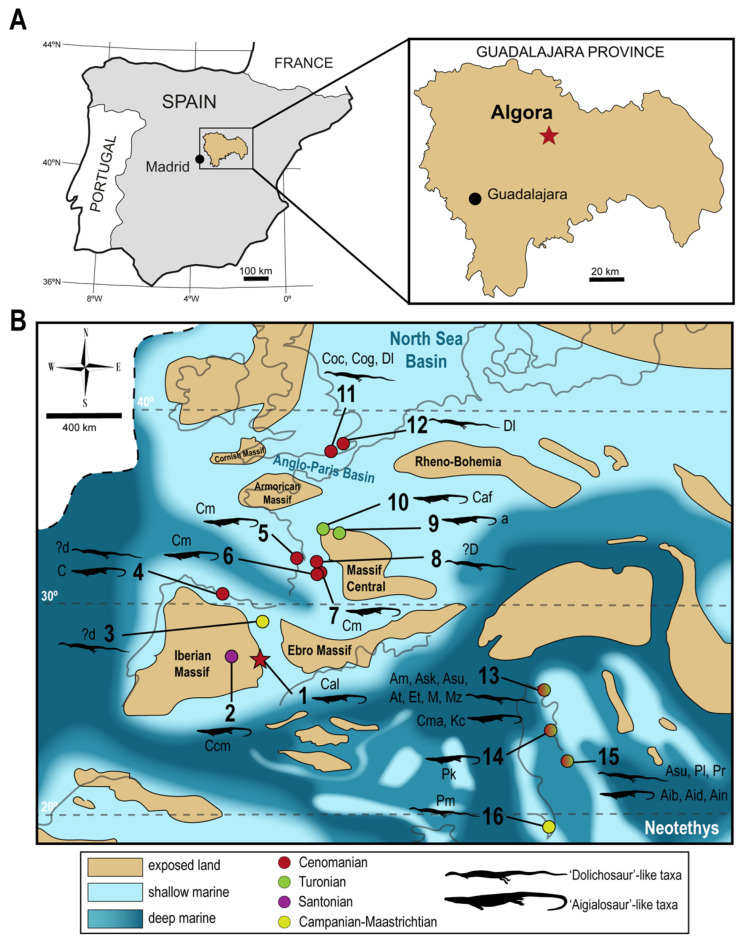
Type locality of *Carentonosaurus algorensis* sp. nov. (Algora, Guadalajara Province, Castilla-La Mancha Autonomous Community, Central Spain) (**A**), and late Cenomanian–early Turonian paleogeographical map (modified from Janetschke and Wilmsen [19]), showing the distribution of marine non-ophidian, non-mosasaurid pythonomorphs (i.e., ‘dolichosaur’-like and ‘aigialosaur’-like taxa, see Discussion) in Europe (**B**). **1**, Algora, Guadalajara (Castilla-La Mancha, Spain) (middle–late Cenomanian); **2**, Pinilla del Valle (Madrid Province, Spain) (Santonian) [20]; **3**, Laño (Burgos Province, Spain) (late Campanian–early Maastrichtian) [12]; **4**, Oviedo (Asturias Province, Spain) (late Cenomanian) [11]; **5**, Madame Island (Charente-Maritime Department, France) (late Cenomanian) [21]; **6**, Angoulême (Charente Department, France) (late Cenomanian) [22]; **7**, La Couronne (Charente-Maritime Department, France) (late Cenomanian) [21]; **8**, La Buzinie (Charente Department, France) (middle early Cenomanian) [22]; **9**, Ferrière-sur-Beaulieu (Indre-et-Loire Department, France) (late Turonian) [23]; **10**, Le Paluau (Indre-et-Loire Department, France) (late Turonian) [24]; **11**, Sussex (England, UK) (early–middle Cenomanian) [25,26,27]; **12**, Kent (England, UK) (early Cenomanian) [26]; **13**, Komen, Škrbina and Tomačevica, (Komen Municipality, Slovenia) (middle Cenomanian–early Turonian) [28,29,30,31,32,33,34,35,36,37,38]; **14**, Savar (Dugi Otok Island, Croatia) (Cenomanian–Turonian) [39]; **15**, Malo Grablje, Starigrad and Vrboska (Hvar Island, Croatia) (late Cenomanian–late Turonian) [29,31,38,40,41,42]; **16**, Nardò (Apulia, Italy) (late Campanian–early Maastrichtian) [13]. a, aigialosaur; Aib, *Aigialosaurus bucchichi*; Aid, *Aigialosaurus dalmaticus*; Ain, *Aigialosaurus novaki*; Am, *Adriosaurus microbrachis*; Ask, *Adriosaurus skrbinensis*; Asu, *Adriosaurus suessi*; At, *Acteosaurus tommasinii*; C, *Carentonosaurus* sp.; Caf, affinis *Carentonosaurus*; Cal, *Carentonosaurus algorensis* sp. nov.; Ccm, *Carentonosaurus* cf. *mineaui*; Cm, *Carentonosaurus mineaui*; Cma, *Carsosaurus marchesetti*; Coc, *Coniasaurus crassidens*; Cog, *Coniasaurus gracilodens*; d, dolichosaur; D, Dolichosauridae; Dl, *Dolichosaurus longicollis*; Et, *Eidolosaurus trauthi*; Kc, *Komensaurus carrolli*; M, *Mesoleptos* sp.; Mz, *Mesoleptos zendrinii*; Pk, *Portunatasaurus krambergeri*; Pl, *Pontosaurus lesinensis*; Pm, *Primitivus manduriensis* Pr, *Pontosaurus ribaguster*.

## 2. Materials and Methods

The material studied here is an isolated vertebra, deposited at the Museo de Paleontología de Castilla-La Mancha (Cuenca, Spain), under the Algora collection number ALG 200. It was collected during the 2021 fieldwork performed at the Algora fossil site (Algora Municipality, Guadalajara Province, Central Spain).

Several measurements were taken for ALG 200 (following Hontecillas et al. [20]), using a 200 × 0.02 mm caliper. Vertebral measurements comprised the following (see Figure 2): prezygapophyseal width (PW), measured as the maximum width between the prezygapophyseal facets in dorsal view; postzygapophyseal width (PtW), measured as the maximum width between the prezygapophyseal facets in dorsal view; distance between paradiapophyses (PD), measured as maximum width across paradiapophyses in dorsal view; neural arch length (NL), measured as maximum length from the anterior margin to posterior margin in dorsal view; centrum length (CL), measured as maximum length from the ventral cotylar rim to the posterior end of the condyle in ventral view; maximum height (MH), measured as the maximum height in lateral view from the highest point of the neural spine to the line on which lowest point of the centrum falls; cotyle width (CW), measured as maximum horizontal length; cotyle height (CH), measured as maximum vertical length; condyle width (CdW), measured as maximum horizontal length; condyle height (CdH), measured as maximum vertical length. Several extinct pythonomorph and squamate taxa were considered for the comparative anatomical study performed here, including, among others, firsthand study of the type material (MNHN IMD 21) of *Carentonosaurus mineaui* (i.e., the sister taxa of that described here). Moreover, the anatomical position of the isolated vertebra within the vertebral series was inferred based on comparisons with extant squamates (i.e., snakes and varanids) and extinct pythonomorphs, as well as following Houssaye et al. [43]. 

The specimen was CT-scanned (with an angle of 108°, so that sections were not completely perpendicular to the axial plane) in the Non-destructive Techniques Laboratory of MNCN in order to obtain non-invasive sections, and also to analyze the inner bone structure and density. The following technical parameters were used: 160 kV, 62 μA, 0.625 thick copper filter, 0.127-pixel size, 900 projections. The image segmentation and visualization of ALG 200 were performed using the Avizo 7.1. software (Lanika Solutions and Visualization Sciences Group (VSG), Bangalore, India), the latter used in the generation of the 3D models (provided as a 3D pdf file in the Appendix A section). The specimen was photographed in dorsal, ventral, cranial and caudal views, along with the generation of the 3D model and schematic drawings indicating the main anatomical structures in those same views. Six virtual thin-sections were considered for the study of the bone compactness (BC): three transverse sections, near the neutral transversal plane (NTP, sensu de Buffrénil et al. [14]); and three longitudinal sections, near the midsagittal plane. These virtual bone sections were processed with Adobe Photoshop^®^ (v.22.0.1) (San Jose, CA, USA) prior to the analysis. Two different software programs (ImageJ^®^ (National Institutes of Health, Bethesda, ML, USA) and BoneProfileR) were used to quantify the BC. The protocol followed to quantify the BC with ImageJ was that used by Houssaye [15], with measurements repeated three times in each section to reduce the potential error and obtain the mean value. The protocol followed to quantify the BC with BoneProfileR was that of Gônet et al. [44]. The global compactness of the vertebra is here considered as the total mean computed on the mean values per slice.

## 3. Systematic Paleontology

Squamata Oppel, 1811 [45]

Anguimorpha Fürbringer, 1900 [46]

Pythonomorpha Cope, 1869 [47]

Genus *Carentonosaurus* Rage and Néraudeau, 2004 [21]

(Figure 3, Figure 4 and Figure 5)

**Figure 3 animals-13-01197-f003:**
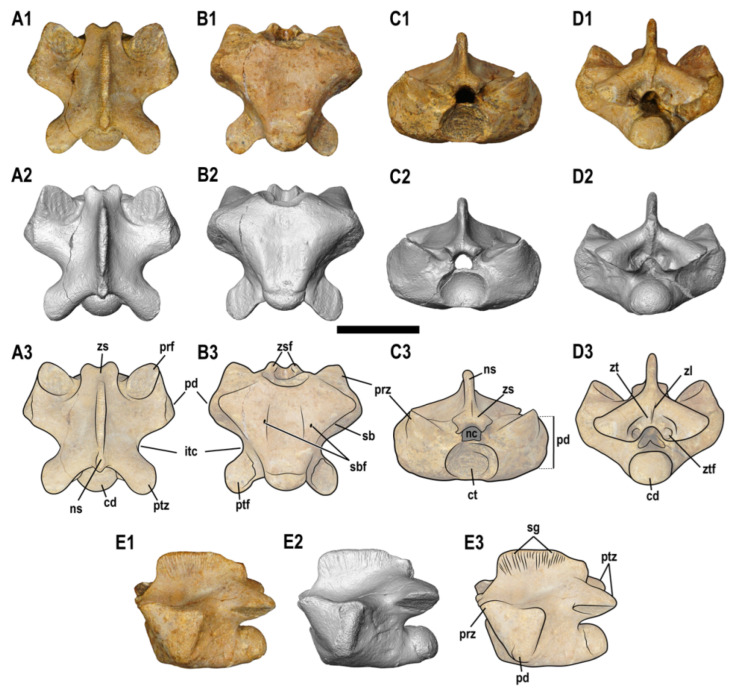
ALG 200, holotype of *Carentonosaurus algorensis* sp. nov., from the uppermost middle to lowermost upper Cenomanian site of Algora (Guadalajara Province, Central Spain). Photo of the original material (**A1**–**E1**), 3D model (**A2**–**E2**), and interpretative drawings on the photos (**A3**–**E3**); in dorsal (**A**), ventral (**B**), cranial (**C**), caudal (**D**), and lateral (**E**) views. Scale bar: 10 mm.

Type species. *Carentonosaurus mineaui* Rage and Néraudeau, 2004 [21].



Included species. *Carentonosaurus mineaui*, *Carentonosaurus algorensis* sp. nov. 

Emended diagnosis (modified from [21]). Member of Pythonomorpha with the following exclusive combination of dorsal vertebral features: neural arch characteristically wider anteriorly (across the prezygapophyses, the paradiapophyses being excluded) than posteriorly (across the postzygapophyses); paradiapophyses strongly extending laterally beyond the prezygapophyses; width across the paradiapophyses notably exceeding that across the postzygapophyses (in dorsal view); extreme pachyosteosclerotic neural arch, with a non-pachyostotic neural spine; prezygapophyseal facets larger than the postzygapophyseal facets; prezygapophyses standing out weakly against the bulk of the paradiapophyses; the condyle clearly exceeding the posterior margin of the neural spine (in lateral view); interzygapophyseal constriction located further back from the mid-length of the neural arch; small, subtriangular neural canal; cotyle wider than the zygosphene; and ‘V’-shaped centrum (in ventral view). Moreover, two additional features can be included in the exclusive combination of the genus (known for its type species, but not for *Carentonosaurus algorensis* sp. nov.): weakly curved ribs, with reduced pseudotuberculum; and dorsoventrally short scapulae and well-defined glenoid areas.

Distribution. Middle Cenomanian to Santonian of Southwestern Europe [11,20,21,22].

**Figure 4 animals-13-01197-f004:**
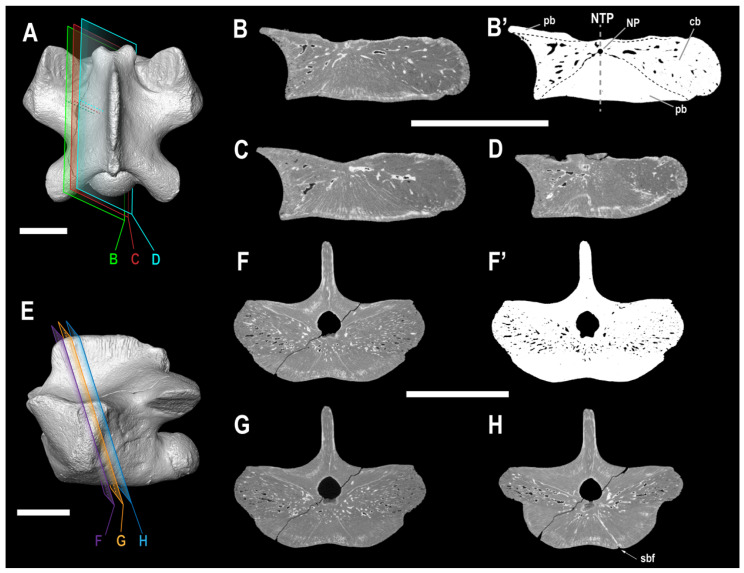
Computed scanning slices of ALG 200, holotype of *Carentonosaurus algorensis* sp. nov., from the uppermost middle to lowermost upper Cenomanian site of Algora (Guadalajara Province, Central Spain). The 3D models show the virtual slices taken for ALG 200 in longitudinal (**A**) and transversal (**E**) sections. The same sections, in the YZ (**B**–**D**) and in the XY (**F**–**H**) axes, were obtained in Avizo 7.1. (**B’**,**F’**), binary transformed images of a representative longitudinal (**B**) and transversal (**F**) section used for the compactness analyses in ImageJ and BoneProfileR. Scale bars: 5 mm (**A**,**E**) and 10 mm (**B**–**D**,**F**–**H**).

*Carentonosaurus mineaui* Rage and Néraudeau, 2004 [21]

(Figure 5A–J,P–Q)



Holotype. MNHN IMD 21, a middle or posterior dorsal vertebra (Figure 5A–E,P).



Referred material. 57 vertebrae (MNHN IMD 1–14; 15–20; 22–50; 53–59; MA MAD 1), one rib (MNHN IMD 51) and a fragmentary pectoral girdle (MNHN IMD 52), all from Madame Island (Charente-Maritime Department, France); three unnumbered vertebrae (see Rage and Néraudeau [21]) from La Couronne (Charente Department, France) [21,22]; and several unnumbered vertebrae (housed at the UR1), from Le Mas and L’Amas (Angoulême, France) [22].

**Figure 5 animals-13-01197-f005:**
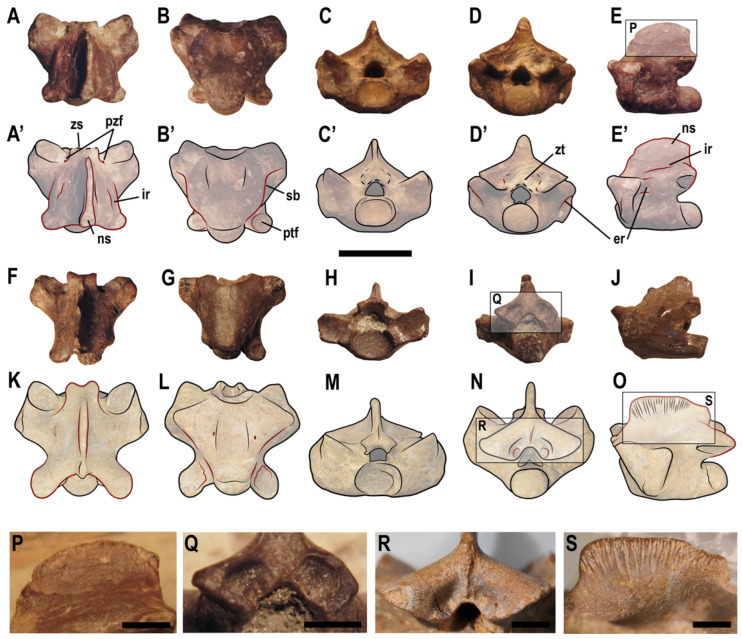
Comparison of vertebral characters in *Carentonosaurus mineaui* (**A**–**J**,**P**–**Q**) and *Carentonosaurus algorensis* sp. nov. (**K**–**O**,**R**–**S**). The red lines indicate characters discussed in the text that differ between *Carentonosaurus mineaui*, here represented by the holotype (MNHN IMD 21) (**A**–**E**,**P**) and a posterior-most cervical vertebra from the type locality (MNHN IMD 40) (**F**–**J**,**Q**), and *Carentonosaurus algorensis* sp. nov. (represented by its holotype, ALG 200) (**K**–**O**,**R**–**S**). Scale bars: 10 mm (**A**–**O**) and 3 mm (**P**–**S**).

Type locality and horizon. Madame Island (Charente-Maritime Department, western France), Dm unit, lower part of the late Cenomanian [21].



Distribution. Upper Cenomanian of western France [21,22].

Emended diagnosis. Member of *Carentonosaurus* differing from *Carentonosaurus algorensis* sp. nov. by the following vertebral features: prezygapophyseal width notably larger than postzygapophyseal width (in dorsal view); epidiapophyseal and interdiapophyseal ridges present; non-strongly laterally projected postzygapophyses; postzygapophyses not (or barely) exceeding the posterior margin of the neural spine (in lateral view); small and rounded postzygapophyseal facets; neural spine with convex and posteriorly inclined anterior margin, sub-convex dorsal margin and straight, tall (relative to the total vertebral height) and thickened posterior margin; non-ornamented neural spine; slightly notched zigosphene, with anterior pointed and parallel lateral borders (concave at the base); shallow concave posterior margin of the neural arch; straight subcentral borders (strongly arched posterolaterally in the dorsals); spine hollowed by a shallow vertical groove; absence of a zygantral lamina; zygantral facets absent.



*Carentonosaurus algorensis* sp. nov.

(Figure 3, Figure 4 and Figure 5K–O,R–S)



Etymology. The specific name is composed of *algor*-, from Algora, the type locality of this taxon, and -*ensis*, a Latin adjectival suffix meaning ‘pertaining to’.

Holotype. ALG 200, an isolated and complete mid-dorsal vertebra (Figure 3, Figure 4 and Figure 5K–O,R–S).

Type locality and horizon. Algora fossil site, Algora Municipality, Guadalajara Province, Castilla-La Mancha Autonomous Community, Castilian Branch of the Iberian Ranges, Central Spain (Figure 1A). Upper part of the Arenas de Utrillas Formation. Uppermost middle to lowermost upper Cenomanian, Late Cretaceous (see Pérez-García et al. [2] and references therein).

Diagnosis. Member of *Carentonosaurus* differing from *Carentonosaurus mineaui* by the following vertebral features: prezygapophyseal width slightly larger than the postzygapophyseal width (in dorsal view); epidiapophyseal and interdiapophyseal ridges absent; strongly laterally projected postzygapophyses; postzygapophyses exceeding the posterior margin of the neural spine (in lateral view); large and ovoid postzygapophyseal facets; neural spine with angled and non-posteriorly inclined anterior margin, sub-straight dorsal margin and sub-convex, low (relative to the total vertebral height) and non-thickened posterior margin; ornamented neural spine; deeply notched zigosphene, with anterior rounded margins and concave lateral borders; deeply concave posterior margin of the neural arch; laterally concave subcentral borders (not strongly arched postero-laterally); neural spine not hollowed by a vertical groove; presence of a zygantral lamina; zygantral facets present.

Anatomical description. ALG 200 is a dorsal vertebra (see Section 4.1), with a total neural arch length of 17 mm and a total centrum length of 13 mm (Table 1). It is a strongly bloated element (i.e., pachyosteosclerotic) (see section *Microanatomical description and bone mass characterization*), being procoelous (Figure 3). The cotyle is oval and deeply concave, slightly wider than high, and wider than the zigosphene in anterior view (Figure 3C). The centrum is cranio-caudally short, and the middle region of its ventral surface is slightly concave (i.e., it displays a slight depression). The condyle is subcircular, being slightly wider than high, and lacking a precondylar constriction in ventral view (Figure 3B). Nevertheless, a clear limit differentiates the outer surface of the centrum body to that of the condyle. In lateral view, the condyle clearly exceeds the posterior margin of the neural spine (Figure 3E). The cotyle–condyle system (sensu Rage and Néraudeau [21]) is oblique, as evidenced by the antero-ventrally faced cotyle (Figure 3B) and the postero-dorsally oriented condyle (Figure 3E). The posterior third of the centrum is partially directed upwards. In ventral view, the centrum is subtriangular (showing a ‘V’-like shape), and it is laterally delimited by blunt subcentral borders, which are notably ‘pinched’ and evenly concave (Figure 3B).

The neural arch of ALG 200 is notably depressed relative to its width (Figure 3C,D). In dorsal view, it is clearly wider anteriorly than posteriorly due to the strong lateral projection of both the paradiapophyses (which extend laterally beyond the prezygapophyses) and the prezygapophyses relative to the postzygapophyses. The neural arch posterior margin is deeply concave (Figure 3A,B). The prezygapophyses are well developed, and they display ovoid articular facets, inclined about 30° relative to the horizontal plane. The prezygapophyses lack any trace of a prezygapophyseal processes (Figure 3A,C) and stand out weakly against the bulk of the paradiapophyses (Figure 3A). The width of the prezygapophyseal facets slightly exceed that of the postzygapophyses (Figure 3A). The postzygapophyses strongly project postero-laterally, exceeding the posterior margins of the condyle and the neural spine (Figure 3E). The postzygapophyseal facets are ventro-laterally directed, defining an angle of about 30° with the horizontal plane. These facets are sub-rounded, being almost equal in size to those of the prezygapophyses (Figure 3B,D). The narrowest part of the interzygapophyseal constriction lies relatively far from the median length of the vertebra, at a position close to the base of the postzygapophyses (Figure 3A3,B3). The paradiapophyses form a single and ellipsoidal articular facet, which slightly bends posteriorly. The paradiapophyses are dorso-ventrally long and antero-posteriorly narrow. They are located anteriorly in the vertebra relative to the centrum length, just behind the prezygapophyses (Figure 3E), and clearly placed above the ventral rim of the cotyle (Figure 3(C3)). In cranial view, the paradiapophyses look bloated. These structures are poorly projected laterally. In ventral aspect, the anterior region of the centrum notably becomes acute towards the lateral borders, connecting with the ventral rim of the paradiapophyses (Figure 3B). The width across the paradiapophyses notably exceeds that across the postzygapophyses. Paired subcentral foramina appear in the anterior half of the centrum, but no parazygosphenal, paracotylar, zygantral or lateral foramina are observed (Figure 3(B3)). Interdiapophyseal and paradiapophyseal ridges are absent in ALG 200 (Figure 3A,E).

The zygosphene is well developed, being recognized as deeply notched in dorsal view (Figure 3A). It bifurcates into two oval and latero-ventrally orientated zygosphenal facets that, together with the neural spine, display an inverted ‘Y’ shape (Figure 3C). The zygosphenal articular facets bear anterior rounded margins, but they are laterally delimited by conspicuous, enlarged and concave borders (Figure 3(A3)). The zygosphene is slightly wider than the neural canal. This canal, which is much lower both in height and width than the cotyle (being about 2.5 times smaller than it) is subtriangular, displaying a ‘D’-like shape, with the convexity facing dorsally (Figure 3C). The zygantrum is formed by two fossae located below the neural spine, with poorly developed zygantral articular facets. A small zygantral lamina longitudinally connects with the base of the neural spine (Figure 3(D2,D3)).

The neural spine is well differentiated from the neural arch, although it remains relatively low along its complete length, being clearly longer than high. In dorsal view, the neural spine is laterally thin and does not extend the entire length of the neural arch, becoming thickest at its middle area (Figure 3A). In cranial and caudal views, the spine width remains constant, such that the posterior margin is not thicker than the anterior one (Figure 3C,D). In lateral view, this spine rises anteriorly as an oblique projection of the zigosphene and then becomes completely perpendicular to the horizontal plane, conforming to an angled anterior margin and reaching its maximum height just behind the posterior margin of the prezygapophyses (Figure 3E). From this point, the height is maintained along most of the spinal length, conforming to a sub-horizontal profile, and the posterior third of the neural spine gradually decreases, establishing a sub-convex and low (relative to the vertebral total height) posterior margin (Figure 3E). The neural spine bears conspicuous longitudinal grooves dorso-laterally located (Figure 3(E3)). It is, unlike the neural arch, not pachyostotic.

Microanatomical description and bone mass characterization. All of the sections examined for ALG 200 show an extremely compact inner osseous configuration, with a mean global compactness of 95.81% (see Table 2, Figure 4). In longitudinal section, two differentiated areas can be recognized: a compact tissue of periosteal origin (i.e., periosteal bone), located on the ventral part of the centrum and on the floor of the neural canal (particularly above the cotyle); and a tissue of endochondral origin (i.e., cancellous bone), located in the remainder of the centrum (Figure 4B,B’). The mean value of the compactness in ALG 200, obtained in longitudinal section is 96.27% (see Table 2). The endochondral tissue is highly compact, with few cavities, and it is displayed as two cones in the longitudinal section, connected by their apices at the neutral point (NP, sensu de Buffrénil et al. [14]). The remainder areas of the centrum are occupied by an extremely compact tissue (i.e., the periosteal bone), which extends from the NP to the ventral region (Figure 4B–D). In transverse section, no trace of a suture between the centrum and neural arch is observed (Figure 4F–H). The mean value of the compactness obtained for the transverse section of this vertebra is 95.34% (see Table 2). ALG 200 shows a hyperplasy (resulting in the volume increase) of the periosteal bone (corresponding to pachyostosis s.s.), as well as a high compact endosteal bone (i.e., osteosclerosis). Due to this last condition, the typical structure observed in the transverse section of the squamate vertebrae, that is, a double-ringed configuration [16,48], is here obscured.

## 4. Results

### 4.1. Anatomical Position of ALG 200 within the Vertebral Column

The vertebral centrum of ALG 200 shows a wide ventral surface and a depressed neural arch relative to its total width (Figure 3B,C). The paradiapophyses are clearly placed above the ventral rim of the cotyle, not reaching the ventral margin of the centrum. However, their ventral margins are slightly placed below the mid-height of the cotyle (Figure 3(C3)). The prezygapophyses and postzygapophyses are well developed. The cotyle and condyle are relatively large, but they are not strongly depressed or small and cylindrical (as in cervical or caudal vertebrae) (Figure 3C,D). Moreover, this vertebra lacks hypapophyses and hypapophyseal peduncles (which are characteristic of the cervical vertebrae), transverse processes (characteristic of sacral and proximal caudal vertebrae) or haemal peduncles and chevrons (characteristic of caudal vertebrae) (Figure 3B). The described combination of characters in this Section 4.1 indicates a dorsal position for ALG 200 in the vertebral series [20,24,43]. Within the dorsal vertebrae, those of the posterior region are characterized by having paradiapophyses located far above the ventral rim of the cotyle (i.e., their ventral margin located at the mid-height of the cotyle or clearly surpassing this region) and distally projected relative to the centrum [43,49]. These features are not present in ALG 200 and, therefore, are recognized as belonging to the anterior to middle dorsal region [43].

### 4.2. Ontogenetic Stage of ALG 200

Typical juvenile features regarding the vertebral anatomy in squamates include, among others: vertebrae with poorly developed articulation areas (e.g., prezygapophyses, postzygapophyses); weak ossifications of both the zygosphene and the zygantrum; weak ossification of the condyle; a marked lateral depression of the cotyle; and a wide neural canal in relation to the overall vertebral size [21,24,50]. ALG 200 lacks any of the afore-mentioned features (Figure 3A–D). Thus, an adult ontogenetic stage is recognized for this specimen. Therefore, the vertebral characters observed in ALG 200 are here considered robust from a taxonomic point of view, not being associated with an early ontogenetic stage.

## 5. Discussion

### 5.1. Systematic Assignment of ALG 200

#### 5.1.1. General Considerations

ALG 200 is a procoelous and non-notochordal dorsal vertebra, this condition being shared with those of the squamates [23,49,51,52,53], in contrast to the commonly amphicoelous and notochordal vertebrae of Rhynchocephalia (the sister clade of Squamata) [51,54]. The general inner vertebral configuration of ALG 200 is concordant with that characteristic of squamates (see Houssaye et al. [15,48]). Within squamates, the obliquity of the axis of the cotyle–condyle system, the posterior position of the narrowest part of the interzygapophyseal constriction, the presence of a well-developed zygosphene–zygantrum complex, and that of a roofed zygosphene, with latero-ventrally directed articular facets, are recognized as an exclusive combination of characters for Pythonomorpha [9,21,24,49,52,53].

Pythonomorpha is composed by Ophidia, Mosasauroidea (i.e., a clade including both Mosasauridae and *Aigialosaurus*), and other taxa whose phylogenetic position within the clade is currently under discussion, some of them being generally referred to Dolichosauridae (sensu Paparella et al. [13]) (see below). In ophidians, the zygosphene is strongly developed and not notched, displaying a particularly straight zygosphene roof in the anterior view; and the zygosphenal articular facets are usually massive and/or widely divergent towards the lateral areas [11,22,24,55,56,57,58,59]. As pointed out by Lee and Scanlon [32] and Palci [60], the presence of vertebral zygapophyses inclined less than 30° may be considered and exclusive character for this clade. In addition, ophidians commonly present prezygapophyseal processes extending laterally from the prezygapophyseal facets [24,49,57,58,59,60,61], and the width across the prezygapophyses is not larger than that across the postzygapophyses [21,24]. Moreover, the dorsal centrum of this clade ventrally displays a low and rounded haemal keel, as well as paired subcentral foramina located in subcentral fossae, limited by subcentral ridges [57,58,59,62]. All of the latter characters are absent in ALG 200 (Figure 3). Therefore, it is not referable to Ophidia within Squamata.

The hydropedal and hydropelvic mosasauroids (i.e., those with flippers and no sacrum), namely the Mosasauridae, developed derived vertebral characters, as they were highly adapted to open-sea environments. Their dorsal vertebrae show the following exclusive combination of derived characters within Squamata: lack of obliquity of the cotyle–condyle system (i.e., the centra are spool-shaped or amphicoelic in ventral view); reduced and/or vertically orientated zygapophyses; and, usually, reduction to absence of zygosphene-zygantrum for at least the middle to posterior dorsals [9,16,20,23,24,29,30,49,52,63,64]. By contrast, the zygosphene–zygantrum complex is retained in primitive mosasaurids (e.g., *Clidastes* or *Ectenosaurus*), which extends to the posterior elements of the dorsal series [9,65], their articular facets being laterally directed (see Houssaye and Bardet [49]; Makádi et al. [66]; and Bardet et al. [67]). All of these features are absent in ALG 200 (Figure 3), not being referable to Mosasauridae.

#### 5.1.2. Comparison of ALG 200 with Non-Mosasaurid, Non-Ophidian Pythonomorphs

The phylogenetic relationships between the non-ophidian, non-mosasaurid pythonomorphs are not agreed upon (see Augusta et al. [68]). While some authors propose the clade Ophidiomorpha (i.e., ‘dolichosaur’-like taxa + Ophidia) [36,37], others consider the clade Mosasauria (i.e., ‘dolichosaur’-like taxa + Mosasauroidea) [13,68,69,70]. Within the non-ophidian, non-mosasaurid pythonomorph representatives, the following can be excluded from further comparisons with the Algora specimen for lacking pachyostotic s.l. vertebrae: *Kaganaias hakusanensis* Evans et al., 2006 [71], from the Valanginian–Hauterivian of Japan [18]; *Aphanizocnemus libanensis* Dal Sasso and Pinna, 1997 [72], from the Cenomanian of Lebanon [18,21]; *Carsosaurus marchesetti* Kornhuber, 1893 [73], from the Cenomanian of Slovenia [18,21,29,30,38]; *Dolichosaurus longicollis* Owen, 1850 [74], from the Cenomanian of England and Germany [18,20,21,26]; *Portunatasaurus krambergeri* Campbell Mekarski et al. [39], from the Cenomanian–Turonian of Croatia; *Coniasaurus* spp., from the Cenomanian–Santonian of England and USA [18,20,21,25,27,75,76,77,78,79], and cf. *Coniasaurus*, from the Albian of Australia [80]. 

Within the ‘dolichosaur’-like pythonomorphs (sensu Bardet et al. [1]), the following taxa have been reported as showing some degree of pachyostosis s.l. in their dorsal vertebrae: *Acteosaurus tommasinii* von Meyer, 1860 [81], from the Cenomanian of Slovenia; *Eidolosaurus trauthi* Nopcsa, 1923 [28], from the Cenomanian of Slovenia; *Adriosaurus* spp, from the Cenomanian–Turonian of Croatia and Slovenia; *Mesoleptos* spp., from the Cenomanian–Turonian of Slovenia; *Pontosaurus* spp., from the Cenomanian–Turonian of Croatia and Lebanon; and *Primitivus manduriensis* Paparella et al. [13], from the Campanian–Maastrichtian of Italy. The dorsal vertebrae of *Acteosaurus tommasinii*, *Eidolosaurus trauthi* and *Adriosaurus* spp. are about 5 mm in length, being much smaller than ALG 200 (see Table 1 in this work and Table 2 in Hontecillas et al. [20]) [20,24,33,35,36,37]. Moreover, *Adriosaurus* spp. differ from ALG 200 in having pachyostotic neural spines [20,21,31]. *Mesoleptos zendrini* Cornalia and Chiozza, 1852 [82], from the Cenomanian–Turonian of Slovenia, was considered as a *nomen dubium* by Rage and Néraudeau [21], and its holotype and referred material are now lost [32,38]. However, some specimens from the Cenomanian of Slovenia and Palestine have been assigned to the genus *Mesoleptos* [18,32,38]. Houssaye [18] reported a “slight” pachyostotic condition only in the anterior dorsal vertebrae of the specimen HUJ-PAL EJ699, from the lower Cenomanian of ‘Ein Yabrud (Palestine), attributed to *Mesoleptos zendrini* (originally referred as to *Mesoleptos* sp. by Lee and Scanlon [32]) and an osteosclerotic condition in another dorsal vertebra (unnumbered specimen) from the same age and locality, also identified as belonging to the same taxon. That osteosclerotic condition recognized for these particular specimens, although Houssaye [18] does not confidently confirm the presence of vertebral pachyostosis for *M. zendrini*, differs from the strong pachyosteosclerotic condition in ALG 200 (see Figure 4). Moreover, the dorsals assigned to the genus *Mesoleptos* display a ‘dolichosaur-like’ type of vertebra, here referred as displaying a ‘Y’-shaped elongated centrum (i.e., paradiapophyses projected anterolaterally, directed at an angle of about 45° to the long axis of the centrum as seen from ventral view [34], the paradiapophyseal width not markedly exceeding that of the postzygapophyses, and the interzygapophyseal constriction more closely placed to the middle portion of the neural arch than to the postzygapophyses (pers. obs.) (see Figure 5 in Vullo et al. [22] and Figure 13 in Houssaye [18]), differing from ALG 200 (Figure 3). *Pontosaurus* spp., were regarded by Rage and Néraudeau [21] as ‘non-pachyostotic’. However, Houssaye [18] indicates a clear pachyosteosclerosis in the dorsal vertebrae of *Pontosaurus lesinensis* Kramberger, 1892 [83], from the Cenomanian of Croatia and *P. kornhuberi* Caldwell, 2006 [84], from the Cenomanian of Lebanon; and *P. ribaguster* Maxine Mekarski, 2017 [38], from the Cenomanian–Turonian of Croatia, is described as having pachyostotic dorsal vertebrae. The dorsal vertebrae of *Pontosaurus* spp. differ from ALG 200 in having a ‘dolichosaur-like’ type of vertebrae (see Maxine Mekarski [38]; Pierce and Caldwell [40]; Caldwell [84]). *Primitivus manduriensis* has also been characterized as having pachyostotic dorsal vertebrae [13], although here considered by the authors as difficult to assess based on the available material of the type and only known specimen. Nevertheless, the dorsal vertebrae of this form are elongate; roughly rectangular in shape; and with cylindrical centra, only slightly anteriorly expanded [13]. This configuration is different than that observed in ALG 200 (Figure 3).

Within the ‘aigialosaur’-like pythonomorphs (sensu Bardet et al. [1]), the following taxa displaying some degree of pachyostosis s.l. in their dorsal vertebrae are known: *Haasiasaurus gittelmani* (Polcyn et al., 1999 [85]) Polcyn et al., 2003 [86], from the Cenomanian of Palestine; *Komensaurus carrolli* Caldwell and Palci, 2007 [34], from the Cenomanian of Slovenia; *Aigialosaurus* spp., from the Cenomanian–Turonian of Croatia; and the afore-mentioned French Cenomanian *Carentonosaurus mineaui*. *Haasiasaurus gittelmani* was recognized by Rage and Néraudeau [21] as ‘non-pachyostotic’, a condition supported by Houssaye [18] for most specimens attributed to it. However, Houssaye [18] identified three anterior dorsal vertebrae in the specimen HUJ-PAL EJ701 and a probably posterior dorsal vertebra (EJ unnumbered), both from the lower Cenomanian of ‘Ein Yabrud (Palestine) and attributed to *Haasiasaurus gittelmani*, as slightly pachyostotic and osteosclerotic, respectively, indicating that this taxon probably had pachyosteosclerotic dorsal vertebrae. The anterior dorsal vertebrae of *H. gittelmani* figured in Houssaye [18] display straight subcentral borders, and the posterior dorsal vertebra of that same taxon figured in Houssaye [16] is of ‘dolichosaur-like’ aspect. Therefore, the vertebral material attributed to this taxon differs from ALG 200 (Figure 3). The dorsal vertebrae of *Komensaurus carrolli* were considered by Houssaye [18] as possibly slightly pachyostotic. They also differ from ALG 200, considering their ‘dolichosaur-like’ aspect [34]. Houssaye [18] reported a ‘possibly slight’ pachyosteosclerosis in the anterior and middle dorsal vertebrae of *Aigilosaurus bucchichi* (Kornhuber, 1901 [87]) Dutchak and Caldwell, 2009 [42], from the Cenomanian–Turonian of Croatia. However, pachyostosis has not been cited or histologically evaluated for *A. dalmaticus* Kramberger, 1892 [83], which was identified as ‘non-pachyostotic’ by Rage and Néraudeau [21]. Nevertheless, the dorsal vertebrae of *A. bucchichi* show a ‘dolichosaur-like’ aspect [29,42,87] and those of *A. dalmaticus* display thick neural spines of a square profile that extend the entire length of the vertebra [29,41], ALG 200 not being compatible with either of these taxa (Figure 3B). The dorsal vertebrae of *Carentonosaurus mineaui* from the Cenomanian of France [21,22] and ALG 200 share the following exclusive combination of characters: neural arches clearly wider anteriorly than posteriorly as a result of the marked lateral projection of the paradiapohyses (which extend laterally beyond the prezygapophyses) (Figure 5A,K); the width across the prezygapophyseal facets exceeds that across the postzygapophyses (Figure 5A,K); prezygapophyseal facets larger than the postzygapophyseal ones (Figure 5A,B,F,G,K,L); dorsal vertebrae with strongly pachyostotic neural arches and non-pachyostotic neural spines (Figure 5A,K); prezygapophyses stand out weakly against the bulk of the paradiapophyses (Figure 5A,K); centrum clearly exceeding the posterior margin of the neural spine, in lateral view (Figure 5E,J,O); small, subtriangular neural canal (Figure 5C,M); anteriorly broadened surface of the centrum (triangular in ventral view), well delimited by subcentral borders (Figure 5B,L); interzygapophyseal constriction further back relative to the mid-length of the neural arch (i.e., being close to the base of the postzygapophyses) (Figure 5A,K); and width across the paradiapophyses notably exceeding that across the postzygapophyses (Figure 5A,K). Rage and Néraudeau [21] and Vullo et al. [22] referred several additional vertebrae (see the ‘referred material’ section) as perhaps belonging to the type species. Because these remains are unpublished (no figurations or descriptions have been documented), they are not included in the list of referred material of *Carentonosaurus mineaui* included here. The specimen from Algora here analyzed and *C. mineaui* (the only species hitherto known for the genus *Carentonosaurus*) share a combination of vertebral features that allow us to recognize them as two closely related forms within the pythonomorphs. Consequently, ALG 200 is here assigned to the genus *Carentonosaurus*.

#### 5.1.3. Systematic Attribution of ALG 200 within the Genus *Carentonosaurus*

The detailed comparison of the Algora specimen with the vertebrae from the type locality (i.e., Madame Island, Charente-Maritime Department, western France) of *Carentonosaurus mineaui* attributed to this taxon, including its holotype (MNHN IMD 21), allows the recognition of a unique combination of characters for ALG 200: postzygapophyseal facets large and ovoid (Figure 5L vs. Figure 5B’); absence of epidiapophyseal and interdiapophyseal ridges (Figure 5K,N,O vs. Figure 5A’,D’,E’); postzygapophyses strongly laterally projected, exceeding the posterior margin of the neural spine in lateral view (Figure 5O vs. Figure 5E’); dorso-ventrally short neural spine (in relation to the total vertebral height), with an angled and non-posteriorly inclined anterior margin, sub-straight dorsal margin and sub-convex, low (relative to the total vertebral height) and non-thickened posterior margin (Figure 5K,O vs. Figure 5A’,E’); conspicuous longitudinal grooves on the neural spine (Figure 5O,S vs. Figure 5E,P); zigosphene deeply notched, with anterior rounded and lateral concave margins (Figure 5K vs. Figure 5A’,F); posterior margin of the neural arch deeply concave (Figure 5K vs. Figure 5A’); subcentral borders laterally concave (Figure 5L vs. Figure 5B’); and presence of zygantral facets and lamina (Figure 5N,R vs. Figure 5I,Q).

Rage and Néraudeau [21] included, as a diagnostic feature of *Carentonosaurus mineaui*, the usual presence of paracotylar, parazygosphenal, and zygantral foramina, but the absence of lateral and subcentral foramina. As has been discussed by several authors, the vertebral foramina are very variable considering both their presence and number within the squamates at the individual level, this fact not being associated with the position within the vertebral column [21,43,49,67,88,89]. The vertebra from Algora only displays subcentral foramina, and no paracotylar, zygantral or parazygosphenal ones are observed, unlike in the dorsals of *C. mineaui* (see above). Therefore, the exclusive presence of subcentral foramina in ALG 200 is probably a difference of systematic value relative to *C. mineaui*. However, due to the reasons mentioned above, and pending for more vertebral material of the taxon from Algora, vertebral foramina were not here considered in the diagnosis of the new Spanish taxon.

Additional vertebral material was assigned to the genus *Carentonosaurus* after the description of the genus: *Carentonosaurus* sp., from the Cenomanian vicinity of the city of Oviedo (Asturias Province), in northern Spain [11]; and *Carentonosaurus* cf. *mineaui*, from the Santonian of Pinilla del Valle (Madrid Province), in central Spain [20]. The material from the Asturias Province (see Vullo et al. [11]) is more similar to the vertebral morphology of *C. mineaui* than that of ALG 200, considering the following combination of characters: prezygapophyseal width notably larger than postzygapophyseal width (in dorsal view); epidiapophyseal and interdiapophyseal ridges present; non-strongly laterally projected postzygapophyses; small and rounded postzygapophyseal facets; and straight subcentral borders, which seem to arch posterolaterally (see Figure 3R–V in Vullo et al. [11]). However, some characters considered in the emended diagnosis of *C. mineaui* cannot be evaluated for the Asturias material (e.g., the morphology of the zygosphene and the neural spine) and, therefore, its attribution to *Carentonosaurus* sp. is supported here. The specimens presented by Hontecillas et al. [20] are significantly similar to the material from the type material of *Carentonosaurus mineaui* (see Rage and Néraudeau [21]), only differing from it in the presence of subcentral and lateral foramina. However, and in agreement with Hontecillas et al. [20] considering the different stratigraphic occurrence (Santonian vs. Cenomanian), the Pinilla del Valle material is referred as *Carentonosaurus* cf. *mineaui*.

Considering all discussed characters, the vertebra from Algora studied here is recognized as more closely related to *Carentonosaurus mineaui* than to any other species of squamate known to date. Although the studied material was restricted to a single element, its excellent preservation, and the fact that isolated vertebrae are considered enough and adequate to make taxonomic determinations within Pythonomorpha [24], this study allowed us the confident assignment of the taxon from Algora to a new species, *Carentonosaurus algorensis* sp. nov.

Houssaye [24] reported material of several vertebrae (including dorsals) from the Turonian of the French locality of Le Paluau, referring them as Pythonomorpha indet. This material is here considered for discussion due to its similarity with the vertebrae of *Carentonosaurus* spp., considering the following combined characters: neural arches of the dorsal vertebrae are characteristically wider anteriorly than posteriorly, both because of the larger prezygapophyseal width in relation to the postzygapophyeal one and because of the marked laterally projected paradiapophyses, the latter strongly extending laterally beyond the prezygapophyses; prezygapophyses stand out weakly against the bulk of the paradiapophyses; the condyle clearly exceeds the posterior margin of the neural spine (in lateral view); the interzygapophyseal constriction is located further back from the mid-length of the neural arch; small, subtriangular neural canal; and ‘V’-shaped centrum (in ventral view). The dorsal vertebrae from the Turonian of Le Paluau differ from that of *Carentonosaurus algorensis* sp. nov. in having sub-straight subcentral borders; a neural spine with a convex anterior and a straight posterior margin; presence of epidiapophyseal ridges; a moderately concave posterior border of the neural arch; moderately projected postzygapophyses; and a zygosphene with sub-straight lateral and acute anterior margins (see Figures 2 and 3 in Houssaye [24]). All of the latter characters are shared with the dorsal vertebrae of *Carentonosaurus mineaui*, but they differ from them in having well-defined zygantral articular facets. Houssaye [24] suggested this material could belong to a new pythonomorph, and as it shares most of the features defined for the genus *Carentonosaurus* but with a combination of vertebral features different from that of *C. mineaui* and *C*. *algorensis* sp. nov., it is here tentatively referred to as aff. *Carentonosaurus*.

### 5.2. Paleoecological and Functional Implications for ALG 200

The dorsal vertebra from Algora identified as the holotype of *Carentonosaurus algorensis* sp. nov. shows an evident bloated aspect (Figure 3), which is indicative of the presence of pachyostosis s.s. The CT-scan sections revealed the extreme compactness of both the periosteal and endosteal tissues of ALG 200, evidencing its pachyosteosclerotic nature (Figure 4). Pachyosteosclerosis is only found in tetrapod taxa secondarily adapted to live in shallow coastal environments [14,15], not being compatible with an efficient terrestrial locomotion [14,15,17,18]. Within Squamata, the pachyostosis s.l. is considered a derived feature for the members of both Pythonomorpha and *Pachyvaranus* [14,15]. The pachyostosis s.l. can be found in a high variety (i.e., in different bones and/or degrees) of combinations, even within a single skeleton (see Houssaye et al. [90] for several scenarios), in amniotes secondarily adapted to an aquatic lifestyle [90,91]. Although specific thresholds have been proposed to objectively determine the degree of both pachyostosis s.s. (e.g., cortical development index higher than 17.7%) and osteosclerosis (e.g., compactness index higher than 81.7%) (see de Buffrenil et al. [92]), the complexity of this issue is higher than previously thought, as has been discussed in subsequent publications [90,91]. Nevertheless, the pachyosteosclerosis reported in the dorsal vertebra of *Carentonosaurus algorensis* sp. nov. suggests a very particular aquatic lifestyle. It has been proposed that the pachyosteosclerosis in the vertebrae results in an augmented vertebral mass, so the locomotor capabilities influenced by inertia (e.g., acceleration or maneuverability) are restrained [14,15,16,17]. Furthermore, the augmented volume between the pachyosteosclerotic vertebrae results in a reduced lateral movement of the axial skeleton (this being the main locomotion among terrestrial and semi-aquatic squamates) and, consequently, in a greater stiffness. This would suggest a sub-undulatory swimming mode for *Carentonosaurus algorensis* sp. nov., as proposed for other mosasauroids and stem-ophidians displaying ‘pachyostosis’ [18]. The inner vertebral configuration of *Carentonosaurus algorensis* sp. nov. is extremely similar to that observed in *C. mineaui*, for which a shallow-aquatic mode of life in near-shore environments has been proposed [15]. No appendicular or girdle elements have been recovered from the taxon of Algora, which would provide highly valuable information for a more accurate approach to its paleoecology. Nonetheless, the microanatomical vertebral features observed in *Carentonosaurus algorensis* sp. nov. are compatible with those observed in the plesiopedal (i.e., those with terrestrial-like limbs) than with the hydropedal and hydropelvic (i.e., those with paddle-like limbs and no sacrum: the Mosasauridae), semi-aquatic or terrestrial pythonomorphs (see Houssaye et al. [15] and Houssaye [16] for comparisons; [91]). Therefore, the taxon from Algora would likely have a similar mode of life to that suggested for *C. mineaui*, that is, as a slow swimmer living near the bottom of near-shore shallow environments [15,18]. This is consistent with the sedimentary environment interpreted for the Algora fossil site (see Pérez-García et al. [2] and reference therein). 

## 6. Conclusions

The present study reports the first record of a squamate from the Cenomanian Algora fossil site (Guadalajara Province, Central Spain), increasing the faunal list of the main locality with remains of Cenomanian vertebrates from southwestern Europe. The combination of vertebral characters displayed by the Algora dorsal vertebra (e.g., obliquity of the axis of the cotyle–condyle; and well-developed zygosphene–zygantrum system, with lateroventrally oriented zygosphenal facets) allows its attribution to Pythonomorpha. Several characters allow its identification as a non-ophidian (e.g., notched zygosphene, pregygapophyseal width larger than postzygapophyseal width, absence of prezygapophyseal processes), and non-mosasaurid (e.g., absence of reduction or of vertical orientation of the zygapophyses in the dorsal region) pythonomorph. A unique combination of vertebral characters in the new material, and its closer phylogenetic relationship to *Carentonosaurus mineaui* than to any other pythonomorph, allows its assignment as a new species attributed to the same genus, *Carentonosaurus algorensis* sp. nov. Consequently, the diagnoses for this genus and for the type and so far only defined species are emended. 

The CT scan of the new pythonormorph from Algora reveals its extreme pachyosteosclerotic condition (incompatible with terrestrial, semi-aquatic or hydrodynamic deep-diving habits), suggesting a very specific mode of life as a slow swimmer of shallow marine environments. This is compatible with the lifestyle proposed for other Cenomanian pythonomorphs, including *Carentonosaurus mineaui*. The description of the taxon defined here, *Carentonosaurus algorensis* sp. nov., provides a new record of a pachyosteosclerotic pythonomorph, highlighting the fast diversification of these shallow marine taxa within a restricted geological (i.e., the Cenomanian) and geographical (i.e., the Mediterranean Tethys) context.

## Figures and Tables

**Figure 2 animals-13-01197-f002:**
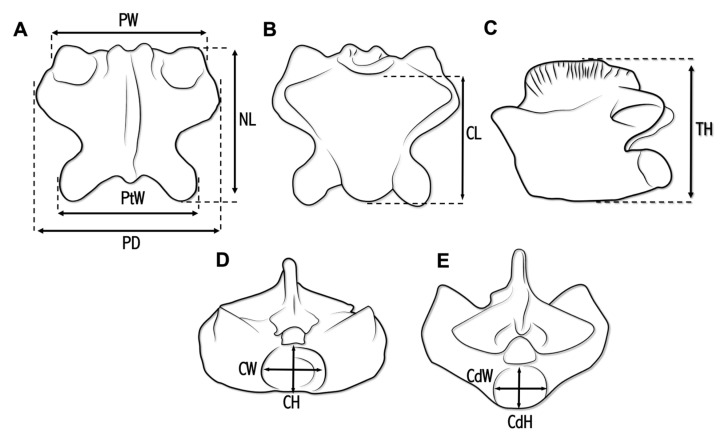
Schematic drawings showing the measurements taken on ALG 200, holotype of *Carentonosaurus algorensis* sp. nov., from the uppermost middle to lowermost upper Cenomanian site of Algora (Guadalajara Province, Central Spain). The vertebra is represented in dorsal (**A**), ventral (**B**), lateral (**C**), cranial (**D**), and caudal (**E**) views.

**Table 1 animals-13-01197-t001:** Vertebral measurements (in mm) taken for *Carentonosaurus algorensis* sp. nov (based on its holotype, ALG 200) and *Carentonosaurus mineaui* (based on its holotype, MNHN IMD 21).

Taxon	PW	PtW	PD	NL	CL	MH	CW	CH	CdW	CdH
*Carentonosaurus algorensis* sp. nov.	16	14.5	20	17	13	14	7	5.5	5	4
*Carentonosaurus mineaui*	11.7 *	10.3	14.1 *	11.2	9.3 *	12.3	5	3.4	4.3 *	3

* Measurements according to [21].

**Table 2 animals-13-01197-t002:** Vertebral compactness, expressed as percentages (%), measured in three transversal (1, 2, 3) and three longitudinal (4, 5, 6) sections of ALG 200, holotype of *Carentonosaurus algorensis* sp. nov., from the uppermost middle to lowermost upper Cenomanian site of Algora (Guadalajara Province, Central Spain).

	1	2	3	4	5	6	Mean
ImageJ	97.11	96.91	96.99	95.94	95.96	97.53	96.74
BoneProfileR	93.80	93.50	93.75	96.00	96.10	96.10	94.88
Section mean	95.46	95.21	95.37	95.97	96.03	96.82	95.81
Mean	95.34			96.27			

## Data Availability

The holotype of *Carentonosaurus algorensis* sp. nov. is deposited in the Museo de Paleontología de Castilla-La Mancha (Cuenca, Spain).

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
