# Peer review of "A New Species of the Pythonomorph Carentonosaurus from the Cenomanian of Algora (Guadalajara, Central Spain)†"

_animals, 2023, doi:10.3390/ani13071197_

Round 1

Reviewer 1 Report

Excellent paper - well written and important. I made minor grammatical suggestions on the manuscript.

Author Response

Point 1: the Reviewer 1 suggest minor changes regarding the spelling and sentence organization.

Response 1: the minor changes suggested by the Reviewer 1 were added to the manuscript, leading to few adjustments or modifications. Thus, several terms and sentences were deleted (“for researchers” (line 10); “are compared took place” (line 42); “so far” (line 50); “the” (line 144); “a” (line 577)), whereas others were replaced (“when” → “between” (line 41); "at the lattermost" → "in the later" (line 72); "corresponds to" → "is" (line 125); “not agreed” → “not agreed upon” (line 420); "to recognize" → "the recognition of" (line 534); "hovering" → "living" (line 641)), following the reviewer recommendations.

Reviewer 2 Report

Cabezuelo-Hernández and Pérez-García describe a new species of Carentonosaurus from the Cenomanian (Upper Cretaceous) of Guadalajara, in Spain. Specifically, the new squamate species is from the site of Algora, which offers one of the few windows on the Cenomanian “European” vertebrate assemblages. The description of the new species is based only on an isolated vertebra, but the inner and outer anatomy of this specimen are discussed in as much details as possible, also including comparisons by direct examination with the type and other material of Carentonosaurus mineaui, providing convincing support for the distinct taxonomic identity of the Algora pythonomorph. The CT-scan investigation also allowed the authors to provide interesting paleoecological considerations.

The text is well-written and focused, and the figures are sharp and effective.

In sum, I think the manuscript is almost acceptable as it is. The only thing I recommend (apart from some minor remarks listed below) is the deposition of a 3D model of the holotype of the new species in a digital repository (and linked as a supplementary file to this article) that would facilitate future comparisons. If the authors are not allowed to share it, a video showing the model rotating in the different norms would suffice.

Should the authors have collected further unpublished measurements on other vertebrae of Carentonosaurus mineaui during their study, it would be nice to include them in table 1.

Perhaps the “simple summary” is not so simple, if it is meant for the general public, in that it uses some taxonomic and geological terms with no explanations (what is a Squamata? How far back in time the Cenomanian goes? Also, the ideas of a European archipelago and the ‘insularity effects’ are challenging to communicate. But, check the journal guidelines, maybe is fine.

Check the English spelling, e.g., palaeo- and -ized

I also listed below (by line) some minor points to check.

L32: I suggest “along the shores”

L45: references plural

L50: call out authorships of taxa at the first mention in the text. Genus species, Author [ref number]

L54: “of which one corresponding … and the other…”

L55: undescribed?

L67: mainly in the Tethys region

L69: , and

L73: developed into; if you wish to stress an evolutionary origin, otherwise use “groups” instead of “lineages”

Figure 1: one scale bar corresponds to 24 km, which seems weird, is there a particular reason?

L98: late…middle…early, sometimes in uppercase, sometimes in lowercase, be consistent throughout the manuscript, where appropriate

L113: Puglia in English is Apulia.

L114: there is a repetition of ;

L147: call out reference.

L150: remove “the” before MNCN.

L151: The following technical parameters were used: …

L230: references plural

L262: Vertebral measurements (in mm), remove Measurements in mm at the end of the caption.

L276: remove both

L338: expressed as percentages

L604: palaeoecological

Author Response

Point 1: I recommend (apart from some minor remarks listedbelow) is the deposition of a 3D model of the holotype of the newspecies in a digital repository (and linked as a supplementary fileto this article) that would facilitate future comparisons. If theauthors are not allowed to share it, a video showing the modelrotating in the different norms would suffice

Response 1: relative to the inclusion of the 3D model as a supplementary file suggested by Reviewer 2, we provide the 3D model as a 3D pdf file (Supplementary file S1).

Point 2: should the authors have collected further unpublishedmeasurements on other vertebrae of Carentonosaurus mineaui during their study, it would be nice to include them in table 1.

Response 2: the Reviewer 2 suggests the inclusion of potential measurements of other vertebrae of Carentonosaurus. The authors do not have additional measurements of other specimen not corresponding to the material of reference (the holotype of Carentonosaurus mineaui).

Point 3: perhaps the “simple summary” is not so simple, if it is meant forthe general public, in that it uses some taxonomic and geologicalterms with no explanations (what is a Squamata? How far backin time the Cenomanian goes? Also, the ideas of a Europeanarchipelago and the ‘insularity effects’ are challenging tocommunicate. But, check the journal guidelines, maybe is fine.

Response 3: the modification of the “Simple summary” section to be less technically expressed has been made, as well as the spelling suggestions following the recommendations suggested by Reviewer 2.

Point 4: the Reviewer 2 suggests several corrections regarding the spelling and some minor points to check.

Response 4: several modifications regarding some terms and expressions were modified considering the suggestions of Reviewer 2: “in the shores” → “along the shores” (line 33); “references” in plural (lines 46, 231); addition of taxon authorships (lines 51-54); “one of them… but the other” → “of which one corresponding … and the other…” (lines 55-56); “undefined” → “undescribed” (line 57); “mainly at the Tethys region” → “mainly in the Tethys region”, addition of comma “, and” (line 70); “several lineages of Pythonomorpha developed ‘pachyostotic’ forms” → “several groups of Pythonomorpha developed into ‘pachyostotic’ forms” (lines 73-74); checking of capital usage of “early”, “late”, “middle” (line 114); “Puglia” → “Apulia” (line 114); double “;;” correction (line 115); reference addition (line 148); “the MNCN” → “MNCN” (line 151); “The following technical parameters are those used in that process” → “The following technical parameters were used:” (lines 152-153); removal of “Measurements” at the end of the caption → “Vertebral measurements (in mm)” (lines 263-265); removal of “both” (line 277); “expressed by percentages” → “expressed as percentages” (line 339); “Palaecological” → “Paleoecological” (line 605) instead of “palaeoecological” as suggested by Reviewer 2, and in consistency with the related terminology throughout the text.

Point 5: Figure 1: one scale bar corresponds to 24 km, which seemsweird, is there a particular reason?

Response 5: required modifications relative to Figure 1 were made according to the Reviewer 2 suggestions: the scale bar has been modified from 24 km to 20 km.